# MELAS-Derived Neurons Functionally Improve by Mitochondrial Transfer from Highly Purified Mesenchymal Stem Cells (REC)

**DOI:** 10.3390/ijms242417186

**Published:** 2023-12-06

**Authors:** Lu Liu, Jiahao Yang, Yoshinori Otani, Takahiro Shiga, Akihiro Yamaguchi, Yasuaki Oda, Miho Hattori, Tsukimi Goto, Shuichi Ishibashi, Yuki Kawashima-Sonoyama, Takaya Ishihara, Yumi Matsuzaki, Wado Akamatsu, Masashi Fujitani, Takeshi Taketani

**Affiliations:** 1Department of Pediatrics, Faculty of Medicine, Shimane University, 89-1 Enya-cho, Izumo 693-8501, Japan; m209432@med.shimane-u.ac.jp (L.L.); jiahao@med.shimane-u.ac.jp (J.Y.); y-oda@med.shimane-u.ac.jp (Y.O.); mhatt001@med.shimane-u.ac.jp (M.H.); t.goto@med.shimane-u.ac.jp (T.G.); yuki.kawashima@med.shimane-u.ac.jp (Y.K.-S.); 2Department of Anatomy and Neuroscience, Faculty of Medicine, Shimane University, 89-1 Enya-cho, Izumo 693-8501, Japan; yotani@med.shimane-u.ac.jp (Y.O.); fujitani@med.shimane-u.ac.jp (M.F.); 3Center for Genomic and Regenerative Medicine, School of Medicine, Juntendo University, Tokyo 113-8421, Japan; t-shiga@juntendo.ac.jp (T.S.); a.yamaguchi.fm@juntendo.ac.jp (A.Y.); awado@juntendo.ac.jp (W.A.); 4Clinical Laboratory Division, Shimane University Hospital, 89-1 Enya-cho, Izumo 693-8501, Japan; 5Department of Digestive and General Surgery, Faculty of Medicine, Shimane University, 89-1 Enya-cho, Izumo 693-8501, Japan; shuichi466@gmail.com; 6Department of Life Science, Faculty of Medicine, Shimane University, 89-1 Enya-cho, Izumo 693-8501, Japan; ishitky@med.shimane-u.ac.jp (T.I.); ymatsuzak@gmail.com (Y.M.)

**Keywords:** MELAS, rapidly expanding clones (RECs), mesenchymal stem cells (MSCs), mitochondrial transfer

## Abstract

Mitochondrial encephalomyopathy, lactic acidosis, and stroke-like episode (MELAS) syndrome, caused by a single base substitution in mitochondrial DNA (m.3243A>G), is one of the most common maternally inherited mitochondrial diseases accompanied by neuronal damage due to defects in the oxidative phosphorylation system. There is no established treatment. Our previous study reported a superior restoration of mitochondrial function and bioenergetics in mitochondria-deficient cells using highly purified mesenchymal stem cells (RECs). However, whether such exogenous mitochondrial donation occurs in mitochondrial disease models and whether it plays a role in the recovery of pathological neuronal functions is unknown. Here, utilizing induced pluripotent stem cells (iPSC), we differentiated neurons with impaired mitochondrial function from patients with MELAS. MELAS neurons and RECs/mesenchymal stem cells (MSCs) were cultured under contact or non-contact conditions. Both RECs and MSCs can donate mitochondria to MELAS neurons, but RECs are more excellent than MSCs for mitochondrial transfer in both systems. In addition, REC-mediated mitochondrial transfer significantly restored mitochondrial function, including mitochondrial membrane potential, ATP/ROS production, intracellular calcium storage, and oxygen consumption rate. Moreover, mitochondrial function was maintained for at least three weeks. Thus, REC-donated exogenous mitochondria might offer a potential therapeutic strategy for treating neurological dysfunction in MELAS.

## 1. Introduction

Mitochondrial diseases (MD) are a group of multisystem disorders that occur when mutations in mitochondria-associated nuclear or mitochondrial DNA (mtDNA) lead to defective oxidative phosphorylation and impaired energy metabolism [1]. In particular, it affects organs with high energy needs, such as the brain [2], where epilepsy, headache, intellectual deficits, and altered mental status are common clinical manifestations [3,4]. The adenine-to-guanine transition (m.3243A>G) at nucleotide 3243 of the mtDNA in the *MT-TL1* gene coding for *tRNAleu^(UUR)^* is one of the most common pathogenic mtDNA mutations that can cause mitochondrial encephalomyopathy, lactic acidosis, and stroke-like episodes (MELAS) [5,6]. MELAS is the most common progressive form of MD and is accompanied by epilepsy, psychopathology, cortical sensory defects, and myopathy [7]. The prevalence of clinically infected individuals harboring the m.3243A>G mutation that causes MELAS is approximately 1:20,000. However, in the general population, the majority of asymptomatic carriers have a rate as high as 1:400 [8,9]. Symptom onset, expression, and clinical severity typically increase with the copy percentage of the m.3243A>G mutation (heteroplasmy) [10,11]. There is no specific treatment for patients with MELAS, and the efficacy of some drugs to enhance mitochondrial residual activity, including vitamin B, Coenzyme Q10, taurine, and arginine, are limited [12,13]. Therefore, a possible strategy for the treatment or palliation of MELAS is imperative. 

The peculiarities of mitochondrial genetics (heteroplasmy phenomena, spontaneous mutations over time) have led to difficulties in modeling mtDNA-associated MD diseases in vivo and in vitro [14]. Most MD cell models do not include the nuclear background of the patient and do not exhibit features of differentiated cells such as post-mitotic neurons, which is not conducive to the study of disease onset and pathogenesis [15,16]. However, human induced pluripotent stem cells (iPSCs) of specific origin overcome these limitations. Numerous studies have shown that disease-specific iPSCs can be differentiated into disease-associated functional cells [17,18,19]. This specific disease cellular model is helpful for understanding the genotypic and phenotypic characteristics of the disease, studying pathogenesis, and providing new ideas for finding effective treatment options and developing new therapeutic drugs. Thus, disease-specific iPSC-derived neurons have significant potential for modeling progressive MDs. In this study, we used iPSCs from patients with MELAS with high levels of heteroplasmy (>60% copy percentage of the m.3243A>G mutation) to establish a MELAS neuron model and evaluate the effects of possible mitochondrial dysfunction on neurons. 

Mesenchymal stem cell (MSC)-based regenerative medicine represents a promising therapeutic strategy. The ease of obtaining MSCs from multiple sources and their low immunogenicity indicate that they can be transplanted into both autologous and allogeneic systems. In a recent study, we reported not only a successful systemic bone regeneration in hypophosphatasia (HPP) by MSC transplantation but also effective suppression of convulsion, one of HPP-related complications [20]. The paracrine, multidirectional differentiation, and mitochondria-targeted transfer capabilities of MSCs have driven translational research and clinical trial evaluations for the treatment of common diseases. In addition, MSCs have been shown to transfer functional mitochondria to recipient cells via multiple pathways, including tunneling nanotubes (TNTs), gap junctions, and microvesicles (MVs) [21]. This exogenous mitochondrial transfer promotes cytoprotection and restores mitochondrial function in various target cells. However, routinely isolated MSCs have high heteroplasmy because of their different proliferation and differentiation functions, leading to therapeutic limitations [22]. In contrast, our previously reported high-purity MSCs (named rapidly expanding clones, RECs) showed superior homogeneity and mitochondrial quality [23,24]. Therefore, this may be a better source of exogenous mitochondria. As an exogenous mitochondrial donor, RECs have a significant restorative effect on bioenergetics and mitochondrial function in mitochondria-deficient cells [25]. However, it is unclear whether RECs donate mitochondria to MELAS neurons and whether these mitochondria are functional or not. 

Therefore, this study aimed to clarify the possible mitochondrial transfer pathway between RECs and neurons and to explore the effect of exogenous mitochondria on mitochondrial function in MELAS neurons, providing a possible strategy for the treatment of MELAS.

## 2. Results

### 2.1. Heteroplasmy Ratio in iPSCs and Neurons Derived from MELAS

We validated the pluripotency of iPS cell lines from healthy donors (control iPSC) and patients with MELAS (MELAS-iPSC^1^, MELAS-iPSC^2^, and MELAS-iPSC^3^). Immunofluorescence staining showed high expression of the pluripotency markers OCT4, TRA-1-60, NANOG, and rBC2LCN in different iPS cell lines (Figure 1A). These data indicate the pluripotency of the individual cell lines. All iPSC lines differentiated into TUJ1-positive excitatory neurons within 18 d from the differentiation start in in vitro (DIV) culturing (Figure 1D). Direct sequencing detected the m.3243A>G mutation in different patient-derived iPS cells, whereas no mutation was present at this locus in the control iPSCs (Figure 1B). We tested whether m.3243A>G heteroplasmy changed with differentiation. While we observed a significant decrease in the level of heteroplasmy (70.6% to 63.8%) after neuronal differentiation, we confirmed a stable level of heteroplasmy during neuronal maturation (>60%; Figure 1C and Appendix A).

### 2.2. High Mitochondrial Transfer Rate of RECs in Contact and Non-Contact Systems

To determine whether RECs and MSCs transferred mitochondria into MELAS neurons, we established contact and non-contact co-culture systems (Figure 2B). In the contact co-culture system, REC and MSC mitochondria were stained with MitoTracker Deep Red, MELAS neuron mitochondria were stained with MitoBright LT Green, and the nuclei were stained with Hoechst 33342. Sequential observations by fluorescence microscopy revealed that at 0 h of co-culture, most cells had independent fluorescence signals, but as the co-culture time increased, some of the MELAS neuron mitochondria with green fluorescence (white arrowheads) fused with red REC- and MSC-derived mitochondria to produce orange fluorescence (Figure 2A). In a non-contact co-culture system, both RECs and MSCs were placed in 0.4 μm (impermeable to mitochondria) and 3 μm (through mitochondria) cell culture inserts, respectively, and after co-culturing for 8 h under the same staining conditions we found that 0.4 μm cell culture inserts blocked the vast majority of mitochondrial translocation, whereas 3 μm cell culture inserts permitted mitochondria to pass through (white arrowheads) and were received by the MELAS neurons (Figure 2C). We further examined the differences in the mitochondrial transfer rates of RECs and MSCs between the two co-culture systems. FCM analysis revealed that the co-culture visualized a DiO/Mito-Red double-positive population (Figure 2D). In the contact co-culture system, there was no significant difference in the mitochondrial transfer rate from RECs and MSCs to MELAS neurons after 4 h of co-culture. In contrast, more mitochondria were transferred from RECs to MELAS neurons after 8 h of co-culturing. The mitochondrial transfer rates at 8 and 24 h were similar, indicating that the mitochondrial transfer efficiency peaked at 8 h of co-culture (Figure 2E). In the non-contact co-culture system, RECs also showed a higher mitochondrial transfer efficiency after 8 h of co-culture (Figure 2F). These data suggest that RECs can transfer mitochondria into MELAS neurons more efficiently than conventional MSCs.

### 2.3. Mitochondrial Transfer via TNTs and Cx43-GJCs

Previous studies have shown that F-actin-mediated TNT formation contributes to the transfer of mitochondria from MSCs to mitochondria-deficient (ρ^0^) cells and restores mitochondrial function in ρ^0^ cells [24,25]. We investigated whether TNTs are involved in mitochondrial transfer between RECs or MSCs and MELAS neurons in vitro. RECs and MSCs (red) were co-cultured with MELAS neurons (blue) for 8 h and then stained with phalloidin (green, selectively labeled F-actin). Fluorescence images showed that TNTs connected RECs and MSCs with MELAS neurons and REC- and MSC-derived mitochondria were also found in TNTs (white arrows, Figure 3A). This suggests a potential role of TNTs in REC- and MSC-mediated mitochondrial transfer. In addition, to verify whether MELAS neurons could receive mitochondria from RECs and MSCs in the absence of TNT formation, we treated cells in the contact co-culture system with cytochalasin D (which inhibits TNT formation without affecting endocytosis) [26], and most TNTs were not formed (Figure 3B,E). Notably, even after the inhibition of TNT formation, some mitochondria (red arrows) were still transferred to the MELAS neurons (Figure 3B). Therefore, we analyzed another possible mitochondrial transfer pathway involving connexin-43-containing gap junction channels (Cx43-GJCs). Studies have shown that MSC-derived mitochondria could be transferred to alveolar epithelial cells via Cx43-GJCs in an in vivo model of lung injury [27]. Our in vitro assays showed high expression of the Cx43 protein between RECs or MSCs and MELAS neurons, suggesting the possible formation of GJC plaques (Figure 3C, white arrows). To verify the specificity of GJCs, we treated the cells with carbenoxelone (a gap junction blocker), which effectively blocked the formation of GJC plaques; however, mitochondria (red arrows) were still transferred into MELAS neurons (Figure 3D). We then analyzed the effects of these two inhibitors on the mitochondrial transfer rates of RECs and MSCs. These results show that gap junction blockers had a greater inhibitory effect on the mitochondrial transfer of RECs (Figure 3F,G). Notably, even with the combination of these two inhibitors, RECs still exhibited a higher mitochondrial transfer rate than MSCs (Figure 3F). These results suggest that there may be other essential mitochondrial transfer pathways for RECs.

### 2.4. Inhibitor Dose Effects on Mitochondrial Transfer

Recent studies have shown that some subtypes of extracellular vesicles (EVs) can carry mitochondrial components from cell to cell (i.e., mitoEVs) and that complete parallel transfer of mitochondria can occur via MVs [28]. Dynasore, an inhibitor of the endocytic pathway, rapidly blocks the formation of coated vesicles, thereby preventing the delivery and reception of mitochondria as MVs. Therefore, we investigated the effect of dynasore on REC- and MSC-derived mitochondrial transfer rates at different doses. The REC- and MSC-donated mitochondrial transfer rate gradually decreased with the incremental dose (Figure 4A), and at a stoichiometry of 80 μM, the mitochondrial inhibition rate reached the threshold, and the REC- and MSC-derived mitochondrial transfer was reduced by 86.2% and 67.1%, respectively (Figure 4B). Carbenoxelone and cytochalasin D also exhibited dose-dependent effects (Figure 4C,E). GJC inhibitor (carbenoxelone) at a stoichiometry of 100 μM resulted in a 66.6% and 45.9% reduction in mitochondrial transfer with RECs and MSCs to MELAS neurons, respectively (Figure 4D). The microtubule/TNT inhibitor (cytochalasin D) at a stoichiometry of 400 nM resulted in 26.5% and 34.8% reduction in mitochondrial transfer with RECs and MSCs, respectively (Figure 4F). The mitochondrial transfer rate of RECs co-cultured with MELAS neurons after treatment with endocytosis inhibitors was significantly lower than that after treatment with GJC inhibitors and microtubule/TNT inhibitors (Figure 4G). This suggests that the endocytosis-mediated mitochondrial transfer pathway is the main pathway involved in REC mitochondrial transfer. In addition, we verified the predominant mitochondrial transfer pathway of RECs in a 3 μm non-contact system (Figure 4H). The results showed that almost no mitochondria were transferred after the addition of endocytosis inhibitors, whereas no significant difference was observed after the addition of microtubule/TNT or GJC inhibitors (Figure 4I). These indicate that the major pathway of mitochondrial transfer of RECs in the 3 μm non-contact system is endocytosis-mediated mitochondrial transfer.

### 2.5. Restoration of the Cellular Bioenergetics via Mitochondrial Transfer

To investigate the effect of RECs and MSCs on the bioenergetics of MELAS neurons, we examined the OCR of each group of neurons. The bioenergetic distribution of neurons in each group is shown in Figure 5A. Coupled and uncoupled respirations were assessed via oligomycin, maximal mitochondrial respiration was determined via FCCP, and non-mitochondrial respiration was assessed via antimycin A and Rotenone (AA/Rot) addition. The results showed that basal respiration, ATP-linked respiration, proton leakage, and maximal respiration were significantly reduced in MELAS neurons compared to control neurons and were substantially restored after co-culture with RECs and MSCs (Figure 5B). To further elucidate whether the improved mitochondrial function was due to the transferred mitochondria, we compared the mitochondrial respiratory function of original MELAS neurons with that of MELAS neurons that received mitochondria (mitochondria isolated and purified from RECs or Rot-treated RECs). Rot has been used to induce mitochondrial dysfunction without affecting cell viability at certain doses [29]. Mitochondria isolated from REC- or Rot-treated (50 nM) RECs were co-cultured with MELAS neurons for 24 h, and the OCR was measured. The cellular bioenergetic profiles of each group are shown in Figure 5C. Compared with MELAS neurons, basal respiration, ATP-linked respiration, proton leakage, and maximal respiration were significantly restored in MELAS neurons with the addition of RECs and Rot-treated REC mitochondria (Figure 5D). Notably, REC mitochondria that were not treated with Rot, restored the mitochondrial respiratory function of MELAS neurons even further (Figure 5D). This demonstrated the benefits of healthy functional mitochondria in recipient cells.

### 2.6. Rescued Mitochondria-Reliant ATP Production

We also examined intracellular ATP and lactate levels and inhibited oxidative phosphorylation (OXPHOS) by adding oligomycin to determine whether the cellular energy source was dependent on OXPHOS. The results showed that in the group without oligomycin, the ATP content of MELAS neurons was significantly reduced, and the lactate content was significantly elevated compared to control neurons, whereas it was significantly restored to the normal level after co-culturing with RECs (Figure 6A,B). After the addition of oligomycin to inhibit OXPHOS, both control neurons and REC- and MSC-treated MELAS neurons exhibited significantly reduced ATP content and significantly elevated lactate content (Figure 6A,B). This suggests that RECs and MSCs improve cellular respiration by enhancing OXPHOS. However, under normal physiological conditions, cells maintain stable cellular ATP levels by altering the rate of ATP production. Therefore, the measurement of total intracellular ATP levels does not provide dynamic information on cellular activity and energy demand. Real-time quantitative analysis of ATP production is a potentially effective method. We assayed mitochondrial OCR and glycolytic extracellular acidification rate (ECAR) using Seahorse XF, which was converted into mitochondrial respiratory, glycolytic, and total ATP production rates. The results showed a 26% decrease, 26% increase, and significant decrease in the rate of ATP production for mitochondrial respiration, glycolysis, and total ATP production, respectively, in MELAS neurons compared to control neurons (Figure 6C). After co-culturing with RECs and MSCs, the ATP production rate of mitochondrial respiration increased by 23% and 13%, respectively, and the ATP production rate of glycolysis decreased by 23% and 13%, respectively, with a significant increase in the total ATP production rate in MELAS neurons (Figure 6C). The ratio of mitochondrial ATP production rate to ATP production from glycolysis was significantly elevated in the co-cultured MELAS neurons, and RECs were superior to MSCs (Figure 6D). These results demonstrated a shift in the mode of mitochondrial respiration from glycolysis to oxidative phosphorylation in treated neurons. In addition, the viability of MELAS neurons was significantly increased after co-culturing with RECs or MSCs (Figure 6E,F).

### 2.7. Recovery of MELAS Neuronal Mitochondrial Function

Mitochondrial reactive oxygen species (ROS) levels were analyzed in each group of neurons using MitoSOX Red staining. This dye is highly selective for mitochondrial superoxide in living cells and produces bright red fluorescence when mitochondrial superoxide is oxidized. Our results showed a significant increase in MitoSOX Red fluorescence in MELAS neurons compared to that in control neurons, and MitoSOX Red fluorescence colocalized with MitoBright LT Green (Figure 7A, white arrows). In addition, there were intense fluorescent spots in the nuclei of some MELAS neurons. Superoxide radicals produced by the nucleus have been reported to cause DNA base modifications and nick formation in the DNA strand, thereby affecting normal physiological processes [30]. Quantification of mitochondrial ROS levels revealed that intracellular ROS levels were significantly reduced in MELAS neurons co-cultured with either RECs or MSCs, and ROS levels in MELAS neurons co-cultured with RECs were significantly lower than those in MELAS neurons co-cultured with MSCs (Figure 7D). Notably, MELAS neurons co-cultured with RECs had a higher mitochondrial content (Figure 7E), and the ratio of MitoSOX Red to MitoBright LT Green was also significantly reduced (Figure 7F). Calcium (Ca^2+^) is a key regulator of mitochondrial function and is involved in mitochondrial ATP production. Excessive accumulation of ROS induces oxidative stress in cells, which in turn increases the cytoplasmic calcium concentration, leading to decreased ATP production and apoptosis [31]. Therefore, we determined the intracellular calcium content by staining with the calcium-sensitive fluorescent dye fluo-8. More fluo-8 fluorescence intensity and spots were observed in MELAS neurons than in control neurons (Figure 7B). Quantification of intracellular calcium concentration showed a significant decrease in MELAS neurons after co-culture (Figure 7G). In addition, intracellular calcium overload and increased ROS can activate redox-sensitive ion channels in the inner mitochondrial membrane, such as the permeability transition pore, inner mitochondrial membrane anion channel, or ATP-sensitive K^+^ channel. The opening of these channels disperses MMP [32]. To determine the MMP level in each group of neurons, we stained each group of neurons using JC-1, a dye whose red-to-green fluorescence intensity ratio decreases when the mitochondria are depolarized. The results showed that MMP was significantly reduced in MELAS neurons, whereas it was significantly restored after co-culture with RECs or MSCs, and that RECs were superior to MSCs (Figure 7C,H). In summary, MMP dynamics were negatively correlated with ROS and elevated intracellular calcium levels. In addition, transmission electron microscopy images showed that both RECs and MSCs significantly ameliorated mitochondrial damage (including cristae deletion, disorder, and breakage) in MELAS neurons (Figure 7I, red arrow). Co-culture with REC or MSC down-regulated the percentage of damaged mitochondria in MELAS neurons (Figure 7J).

### 2.8. Retention of Mitochondrial Function

To determine whether the mitochondrial function of MELAS neurons restored by REC-transferred mitochondria is stable and sustained, neurons differentiated from iPSC in patients with MELAS with or without co-culturing with RECs for 24 h were tested for neuronal mitochondrial function at 1, 7, 14, and 21 d, as shown in Figure 8A. The results showed that compared to MELAS neurons, co-cultured MELAS neurons maintained calcium (Figure 8B,C) and MMP homeostasis (Figure 8D,E) for at least 21 d. Notably, several studies have indicated that GDF-15 is both a diagnostic and severity marker of MD [33,34]. However, the relationship between our therapeutic interventions and GDF-15 levels remains unclear. Our results showed that the mean level of GDF-15 in the MELAS neuron medium was 1148.6 ± 366.2 pg/mL, which was significantly higher than that of the control neurons, which was 294.9 ± 51.1 pg/mL. The levels of GDF-15 after REC and MSC interventions were 267.9 ± 39.5 pg/mL and 414.1 ± 55.7 pg/mL, respectively (Figure 8F). The maintenance of GDF-15 levels by RECs lasted for at least 21 d (Figure 8G). In addition, REC mitochondria showed stable sustainability in maintaining the bioenergetics of MELAS neurons. After co-culturing with RECs, MELAS neurons maintained a high rate of mitochondrial ATP production for at least 21 d, and oxidative phosphorylation was the main mode of energy acquisition (Figure 8H,I).

## 3. Discussion

Our previous report revealed that exogenous REC-donated mitochondria can be transferred into mitochondria-deficient (ρ^0^) cells and restore their mtDNA content and mitochondrial function through multiple pathways [24,25]. However, the role of exogenous mitochondria in specific MD and related neurological functions remains unclear. In this study, we used patient-derived iPSCs with high MD heteroplasmy (>60%, m.3243A>G) to provide a possible neuronal model (MELAS neurons). We found that MELAS neurons exhibited the morphology of mature neurons but with destabilized mitochondrial function. Notably, mitochondria from RECs can be transferred into MELAS neurons via the classical pathway, providing multiple benefits, including enhanced cellular bioenergetics, restoration of respiratory function, and OXPHOS-dependent cellular growth, and this functional restoration is sustained. These findings suggest that the transfer of exogenous mitochondria can reestablish normal physiological functions associated with healthy mitochondria.

Human mtDNA encodes many key proteins involved in the assembly and activity of the mitochondrial respiratory complex and is closely associated with MD [35]. The mutation rate of mtDNA is high because of the lack of histones and effective repair mechanisms in its structure [36]. The percentage of mutated copies of mtDNA (heteroplasmy) plays a role in the development of symptoms, as well as the severity of the disease, with high heteroplasmy (>60%) usually accompanying disease pathogenicity and phenotypic manifestations. Due to the uniqueness and complexity of mtDNA, the establishment of a stable cellular model of MD is fundamental to this study [10]. Notably, human iPSCs appear to be a powerful tool for disease modeling, and because of the natural heteroplasmy of the primitive fibroblast population accompanied by variations in respiratory chain activity, iPSC reprogramming produces clones with varying levels of heteroplasmy [37]. This allowed us to use neurons with appropriate levels of heteroplasmy and respiratory function for disease modeling. In addition, the m.3243A>G mutation (high level of heteroplasmy), which is strongly associated with MELAS, may affect mitochondrial protein synthesis by decreasing the efficiency of amino acid binding during the translation of the 13 mtDNA-encoded proteins in OXPHOS subcomplexes I-V [38]. OXPHOS complex (I-V) deficiency usually leads to an imbalance in the cellular redox state accompanied by increased ROS production and mitochondrial damage [39]. Our study showed that MELAS neurons carrying a high level of heteroplasmy (m.3243A>G mutation) faithfully replicated the features of respiratory complex deficiency, including high levels of ROS, OXPHOS defects, respiratory dysfunction, and decreased cell proliferation. In addition, the dependence on anaerobic glycolysis and energy deficiency exhibited by MELAS neurons provides clues for understanding the pathological mechanisms associated with abnormal energy metabolism in the brain.

MSCs have been shown to repair damaged cells through paracrine effects, such as the release of immunomodulatory factors, extracellular vesicles, microRNAs, and mitochondrial transfer [22]. Multiple mitochondrial transfer mechanisms can restore the bioenergetic requirements of damaged cells [21]. TNTs, a classical mitochondrial transfer pathway, consist of a spontaneous tubular membrane protrusion with an extension of the plasma membrane that permits uni- or bi-directional transport of a wide range of cellular components or organelles, including mitochondria [40]. This is consistent with our finding of mitochondrial transfer in TNTs constructed between RECs and MELAS neurons. Notably, RECs exhibited a superior mitochondrial transfer rate to MSCs in both our established contact and non-contact co-culture systems; however, REC-mediated transfer produced a lower number of TNTs than MSC-mediated transfer. Furthermore, mitochondria from donor cells remained present in MELAS neurons even after the inhibition of TNT formation using the actin polymerization inhibitor cytochalasin D, suggesting the possibility and differences in mitochondrial transfer from RECs and MSCs to MELAS neurons via multiple pathways. The outward growth and length of TNTs depend on the involvement of F-actin, which has bending resistance properties [41]. In contrast, cells with low motility usually accumulate a large number of stress fibers, with F-actin as the main component, accompanied by an increase in cell size [42,43]. Our previous report showed that RECs have a small morphology, high migration and proliferation properties, and lower F-actin expression than MSCs [23]. This explains the low formation of TNTs by RECs and suggests that TNT-mediated mitochondrial transfer may not be predominant in RECs. In addition, it has been reported that MSCs and alveolar epithelium restore alveolar bioenergetics by forming Cx43-containing GJCs to release mitochondria-containing MVs [27]. Our study found high expression of Cx43 at cell junctions, suggesting the possible formation of GJC plaques. In contrast, when the cell junction blocker carbenoxelone was added, Cx43 protein expression was reduced (disappearance of gap junction plaques), and REC and MSC mitochondrial metastases were significantly reduced, suggesting a potential role for Cx43-GJCs in REC mitochondrial metastasis. However, even if both mitochondrial transfer pathways, TNTs and GJCs, are blocked, donor cell mitochondria still exist in MELAS neurons. EVs are essential intercellular communication vectors for the transfer of bioactive substances between cells and organs. Recently, abundant evidence has shown that mitochondria-derived vesicles (MDVs, ~70–150 nm) or intact mitochondrial translocation is mainly mediated by multivesicular bodies (MVBs, ~30–150 nm) and MVs (~200–1000 nm) and that vesicle formation as well as release cannot be separated from the role of endocytosis in cells [28]. We used dynasore to inhibit the formation and release of MVBs and MVs, thereby inhibiting mitochondrial transfer and reception. The mitochondrial transfer rate of RECs was significantly reduced compared to that of MSCs, showing the strongest inhibition efficiency among multiple inhibitions. This revealed the importance of EVs in the mitochondrial transfer mechanism of RECs.

Furthermore, mitochondrial replacement therapy (MRT) of oocytes or fertilized eggs, such as prokaryotic (PNT), spindle (ST), or polar body (PBT) transfer, prevents the second-generation transmission of mtDNA defects. However, germline gene therapy involves the permanent correction of mutated genes in germ cells, which can result in the transmission of alterations to the offspring; therefore, ethical issues, as well as social and legal barriers to the application of MRT in clinical practice, remain [44]. Because MSCs can donate mitochondria to recipient cells, they offer potential possibilities for the treatment of MD. However, there are limitations to the treatment with MSCs. MSCs isolated from human BM using traditional methods proliferate and differentiate inconsistently, the cell populations obtained are highly heterogeneous, and autologous BM-derived MSCs are expensive [22]. In addition, the collection of adult BM-derived MSCs is highly invasive and carries a high risk of infection. In contrast, RECs have low batch-to-batch variability, uniform cell size, high proliferation rate and mitochondrial content, and no ethical issues [23,24]. Our results also showed that REC is advantageous in restoring mitochondrial functions, such as elevated MMP, ATP, OCR, and the regulation of ROS homeostasis. These findings highlight the advantages of functional mitochondria (RECs) in restoring neuronal mitochondrial function (Figure 9).

This study had some limitations. Our observations were based entirely on in vitro experiments, and neuropathophysiological correlations should be confirmed through in vivo experiments. Second, although this study revealed the importance of EVs in REC mitochondrial transfer, this was only initially verified laterally through inhibition experiments, and the specific mechanism of EV-mediated mitochondrial transfer needs to be further elucidated.

## 4. Materials and Methods

### 4.1. Culture of Undifferentiated iPSCs

Human control iPSC lines (201B7) and MELAS-iPSC lines (MELAS-iPSC^1^, MELAS-iPSC^2^, and MELAS-iPSC^3^) were obtained from RIKEN CELL BANK (Ibaraki, Japan) and cultured in an undifferentiated state without MEF feeder cells in StemFit AK02N medium (AJINOMOTO, Tokyo, Japan). The iPSCs were dissociated into single cells with TrypLE select (Invitrogen, Carlsbad, CA, USA) and reseeded at a density of 2 × 10^4^ cells per well on an iMatrix511- treated (Nippi, Tokyo, Japan) 6-well plate with StemFit AK02N containing Y27632 (Fijifilm/WAKO, Osaka, Japan). The plate was incubated in an atmosphere containing 5% CO_2_ at 37 °C. The day after passage, the medium was replaced with StemFit AK02N without Y27632. Afterward, medium replacement was carried out every 2–3 days.

### 4.2. In Vitro Neuronal Differentiation of iPSCs

The procedure of in vitro neural differentiation was performed as described previously [45]. In brief, all iPSCs were cultured in StemFit AK02N medium supplemented with three inhibitors (3i) (3 μM CHIR-99021 (REPROCELL, Kanagawa, Japan), 3 μM SB-431542 (Fijifilm/WAKO), 3 μM dorsomorphin (Sigma, Carlsbad, CA, USA)) from 3 d after passages. The StemFit AK02N with 3i medium was changed daily. On day 5, after the 3i treatment, iPSC colonies were enzymatically dissociated into single cells using TrypLE Select. The dissociated cells were cultured in a suspension at a density of 1 × 10^5^ cells/mL in culture dishes with neuronal induction medium consisting of media hormone mix (MHM) medium (KBM Neural Stem Cell Kit, KOHJIN BIO, Saitama, Japan) supplemented with 2% B27-Minus vitamin A (Invitrogen), 20 ng/mL bFGF, (Peprotech, Tokyo, Japan), and 2 μM SB431542 in a hypoxic incubator (4% O_2_; 5% CO_2_). Three days after neural induction, 3 μM CHIR99021 and 3 μM purmorphamine (Millipore, Darmstadt, Germany) were added. iPSCs were used to form primary neurospheres (NSs) approximately seven days after neural induction. To prepare secondary and tertiary NSs, primary NSs were repeatedly enzymatically dissociated into single cells and suspended in the same neural induction medium.

For neuronal cell differentiation, dissociated tertiary NSs were plated onto coverslips, 12 mm in diameter, coated with 0.1 mg/mL poly-L-ornithine (PLO, Sigma-Aldrich, St. Louis, MO, USA) and 10 ug/mL hFibronectin (R&D Systems, Minneapolis, MN, USA). These cells were cultured in differentiation medium consisting of B-27^TM^ Plus Neuronal Culture System (Gibco^TM^, New York, NY, USA), 10 uM DAPT (FIjifilm/WAKO), 10 ng/mL recombinant human GDNF (FIjifilm/WAKO), 10 ng/mL recombinant human BDNF (FIjifilm/WAKO), and 200 μM ascorbic acid (Sigma-Aldrich, St. Louis, MO, USA) for 7–28 d in a humidified atmosphere containing 5% CO_2_ at 37 °C. Half of the medium was changed every 2–3 days.

### 4.3. Mesenchymal Stem Cells (MSCs) and Rapidly Expanded Clones (RECs)

The bone marrow (BM)-derived MSCs were purchased from Lonza (Basel, Switzerland). RECs were purchased from PuREC Co., Ltd. (Izumo, Japan). Three different RECs and MSCs clones were prepared separately. RECs and MSCs were cultured in DMEM/F-12 medium (FIjifilm/WAKO) supplemented with 15% HyClone fetal bovine serum (FBS, Cytiva, Tokyo, Japan), 1% L-Alany-L-Glutamine Solution (FIjifilm/WAKO), 1% penicillin–streptomycin (FIjifilm/WAKO), 0.25mM L-Ascorbic Acid Phosphate Magnesium Salt n-Hydrate (FIjifilm/WAKO), and 10 ng/mL basic fibroblast growth factor (bFGF, AJINOMOTO, Tokyo, Japan) in a humidified atmosphere containing 5% CO_2_ at 37 °C, until 80% confluent. The medium was changed every 2–3 days. MSCs and RECs from passage 4 were used for further experiments.

### 4.4. Immunofluorescence

Cells in each group were fixed with 4% paraformaldehyde (PFA) for 10 min, washed with PBS, permeabilized with 0.1% Triton^TM^ X-100 (Fijifilm/WAKO, Osaka, Japan)/PBS solution for 10 min, and blocked with blocking buffer (0.1% BSA/PBS) for 1 h. Blocking solution was aspirated and the cells were incubated overnight at 4 °C with diluted primary antibodies (Nanog (D73G4) XP^®^ Rabbit mAb (1:200, Lot:4903P, Cell Signaling Technology, Danvers, MA, USA), Oct-4A (C30A3) Rabbit mAb (1:500, Lot:2840S, Cell Signaling Technology), TRA-1-60(S) (TRA-1-60(S)) Mouse mAb (1:150, Lot:4746S, Cell Signaling Technology), and Connexin 43 rabbit mAb (1:1000, Lot:83649S, Cell Signaling Technology)). Cells were washed with PBS and stained with goat anti-mouse IgG Alexa Fluor^TM^ 488 (1:1000, Lot: A21042, Invitrogen), goat anti-rabbit IgG Alexa Fluor^TM^ 555 (1:1000, Lot: A27039, Invitrogen), and goat anti-rabbit IgG Alexa Fluor 488 (1:500, Lot: A27034, Invitrogen) antibodies for 1 h at room temperature (RT). After washing twice with PBS, the cells were mounted with 1 μg/mL Hoechst 33342 (Invitrogen, Carlsbad, CA, USA). Immunofluorescent staining results were visualized using a BZ-X710 microscope (BZ-X810; KEYENCE, Osaka, Japan).

### 4.5. m.3243A>G mutation Analysis

Genomic DNA was extracted from each cell group using a QIAamp DNA Micro Kit (Qiagen, Hilden, Germany). The forward and reverse primers were -GGACAAGAGAAATAAGGCC- (m.3130-3149) and -AACGTTGGGGCCTTTGCGTA- (m.3130-3149). To determine the presence of the m.3243A>G mutation in iPSCs derived from patients with MELAS. PCR amplification products from each cell set were then sequenced (Applied Biosystems). The level of heteroplasmy for the m.3243A>G mutation was determined according to a previously described method using PCR–restriction fragment length polymorphism (RFLP) [46]. Briefly, using the primers shown above, the wild-type mtDNA amplified a fragment of 294 bp. In the presence of the m.3243A>G mutation, the PCR product was cleaved by ApaI (Thermo Fisher Scientific, Vilnius, Lithuania) restriction endonuclease into two fragments of 178 and 116 bp. The ApaI-digested PCR product was electrophoresed on 2% agarose gel. The proportion of the m.3243A>G heteroplasmy (%) could be calculated by analyzing the electrophoretic bands using the formula: Proportion of mutant = mutant band density/(mutant band density + wild-type band density) × 100%.

### 4.6. Direct Contact Co-Culture System

Before co-culturing, RECs and MSCs were labeled with 200 nM MitoTracker Deep Red (Invitrogen, Carlsbad, CA, USA) and 1 μg/mL Hoechst 33342 (Invitrogen, Carlsbad, CA, USA) for 20 min at 37 °C. MELAS neuron cells were labeled with 0.1 μmol/L MitoBright LT-Green (Dojindo, Kumamoto, Japan) and 1 μg/mL Hoechst 33342 for 15 min at 37 °C. The cells were co-cultured in a B-27^TM^ Plus Neuronal Culture System (Gibco^TM^, New York, NY, USA) supplemented with 1% L-glutamine and 1% penicillin–streptomycin. Cell viability was determined using a 4% trypan blue solution, and the results were frequently greater than 95%. After 0, 2, 4, 8, and 24 h of co-culture, the mitochondrial fluorescence transfer of RECs and MSCs was observed by fluorescence microscopy (BZ-X810, KEYENCE, Osaka, Japan). Flow cytometry (CytoFLEX, BECKMAN COULTER, Indianapolis, IN, USA) was performed to analyze the mitochondrial reception of RECs and MSCs by MELAS neurons. After sorting of 5 μL/mL Vybran^TM^ DiO-labeled MELAS neuron cells (Green, Invitrogen, Carlsbad, CA, USA) and MitoTracker Deep Red-labeled RECs/MSCs, mitochondrial transfer rates were calculated for the Q2 phase (double-positive) distribution as a percentage of total MELAS neuron cells and normalized to the data. The data were analyzed using FlowJo^TM^ software (Version 10, BD, Ashland, OR, USA).

### 4.7. Non-Contact Co-Culture System

After labeling REC/MSC mitochondria with 200 nM MitoTracker Deep Red (Invitrogen, Carlsbad, CA, USA), REC/MSC were collected and placed in 0.4 μm or 3 μm cell culture inserts for 4 h of incubation. Neuronal cells of each group were labeled with 0.1 μmol/L MitoBright LT-Green mitochondrial labeling solution (Dojindo, Kumamoto, Japan) and 1 μg/mL Hoechst 33342 (Invitrogen, Carlsbad, CA, USA), inoculated in 24-well plates, and co-cultured with REC/MSC in 0.4 μm or 3 μm cell culture inserts.

### 4.8. Microscopic Image

To observe possible mitochondrial transfer pathways (microtubules/TNTs) of RECs or MSCs in a direct co-culture system, mitochondria of REC/MSC were stained using 200 nM MitoTracker Deep Red (Invitrogen, Carlsbad, CA, USA) at 37 °C for 20 min before co-culture, and 1 μg/mL Hoechst 33342 (Invitrogen) was used to stain neuronal cells of each group for 15 min at RT. After two washes with PBS, the cells were trypsinized and co-cultured for 6 h in 24-well plates. The cells were fixed with 4% paraformaldehyde (PFA) in PBS at RT for 15 min and permeabilized with 0.1% Triton^TM^ X-100 in PBS for 5 min. Microtubules/TNTs were detected by staining the fixed cells with 1X Green Fluorescent Phalloidin Conjugate working solution (Abcam, ab112125, Cambridge, UK) for 45 min at RT. After gentle washing twice with PBS, the cells were visualized under a fluorescence microscope (BZ-X810; KEYENCE, Osaka, Japan).

### 4.9. Drug Dose Dependency

Analysis of the effects of different dose gradients of different inhibitor compounds on the donor mitochondria of RECs and MSCs. RECs and MSCs were stained with 200 nM MitoTracker Deep Red (Invitrogen, Carlsbad, CA, USA) before co-culture. MELAS neurons were stained with 5 μL/mL Vybrant^TM^ DiO. Subsequently, the inhibitory compounds dynasore (Biovision, Milpitas, CA, USA), carbenoxelone (Sigma-Aldrich, Saint Louis, MO, USA), and cytochalasin D (Sigma-Aldrich, Saint Louis, MO, USA) were added separately during co-culture. After co-culturing with CytoFLEX (BECKMAN COULTER, Indianapolis, IN, USA), the Q2 phase (double-positive) distributions were calculated as a percentage of the total number of MELAS neuron cells. The data were analyzed using FlowJo^TM^ software (Version 10, BD, Ashland, OR, USA).

### 4.10. Transmission Electron Microscopy

Neuronal cells (2 × 10^5^) from each group were seeded in 35 mm dishes. After 24 h, cells were pre-fixed in 2.5% glutaraldehyde electron microscopy solution (FIjifilm/WAKO), 2% PFA, and 0.1 M phosphate buffer (FIjifilm/WAKO) for 2 h at RT and washed thrice with washing buffer. Fixed cells were dehydrated sequentially in ethanol (50%, 70%, 90%, 95%, and 100%), then infiltrated and embedded in epoxy resin before left to harden at 60 °C for 24 h. Ultrathin sections were obtained using a diamond knife and copper grids (400 mesh; NISSHIN EM, Tokyo, Japan). Sections were stained with uranyl acetate, and neuronal mitochondria from each group were visualized using a transmission electron microscope (Topcon EB-002B, Tokyo, Japan).

### 4.11. Measurement of Mitochondrial Membrane Potential (MMP)

After 8 h of non-contact co-culture, the MMP of the different groups was assessed using a JC-1 Detection Kit (Dojindo, Kumamoto, Japan), which exhibits potential fluorescence characteristic changes in the mitochondria. The red/green fluorescence intensity ratio of JC-1 decreased in depolarized mitochondria owing to the disruption of red fluorescent J-aggregates. Briefly, 7 × 10^4^ cells/mL were incubated with 1 mmol/L JC-1 working solution for 40 min at 37 °C. The supernatant was discarded, and cells were washed twice with HBSS (Gibco^TM^; Paisley, UK). An imaging buffer solution was added, and the cells were observed under a fluorescence microscope (BZ-X810; KEYENCE, Osaka, Japan). The ratio of mitochondrial JC-1 red (590 nm) to green (530 nm) was considered representative of cell MMP. Cellular MMP for each group was detected using a GloMax^®^ Discover Microplate Reader (Promega, WI, USA) based on the red-to-green fluorescence intensity ratio.

### 4.12. Measurement of Reactive Oxygen Species (ROS)

MitoSOX^TM^ Red fluorescent probe (Life Technologies, Carlsbad, CA, USA) was used to visualize mitochondrial superoxide production according to the manufacturer’s protocol. Briefly, cells (1 × 10^5^) grown on 24-well plates were washed twice with PBS to remove the medium and incubated with 5 μM MitoSOX working solution for 10 min at 37 °C. After three gentle washes with warm buffer (HBSS; Gibco, Paisley, UK), the cells were imaged immediately under a fluorescence microscope (BZ-X810; KEYENCE, Osaka, Japan). To confirm the mitochondrial localization of MitoSOX, cells were labeled with 0.1 μmol/L MitoBright LT-Green (Dojindo, Kumamoto, Japan) for 15 min at 37 °C. The mean fluorescence intensities of MitoSOX-Red and MitoBright LT-Green were determined using a GloMax^®^ Discover Microplate Reader (Promega).

### 4.13. Detection of Intracellular Calcium

Intracellular calcium levels were measured in each group using the Fluo-8 Calcium Flux Assay Kit (Abcam, ab112129). In brief, cells were incubated with Fluo-8 for 30 min at 37 °C and 30 min at RT in HHBS buffer (Sigma-Aldrich, St. Louis, MO, USA), according to the manufacturer’s instructions. Fluorescence was measured using a GloMax^®^ Discover Microplate Reader (Promega) with a filter set of Ex/Em = 490/525 nm. The calcium fold change was calculated using no stimulation data as the standard value.

### 4.14. Seahorse XF Analysis

A Seahorse XF HS mini analyzer (Seahorse Bioscience, Agilent, CA, USA) was used to assess the key parameters of mitochondrial function by directly measuring the cellular respiratory index of oxygen consumption rate (OCR) and the glycolysis index of extracellular acidification rate (ECAR). According to the manufacturer’s protocol, cells in each group were seeded onto an XFp cell culture microplate (Seahorse Bioscience, Agilent, Santa Clara, CA, USA) at a density of 30,000 cells per well for 24 h. On the day of the experiment, the culture medium was replaced with Seahorse XF RPMI medium (Seahorse Bioscience, Agilent, CA, USA) supplemented with 10 mM glucose, 2 mM l-glutamine, and 1 mM sodium pyruvate (pH 7.4) and transferred to a non-CO_2_ incubator for 60 min. The final concentration used for mito stress assay (Agilent, CA, USA) was processed by sequential addition of 1.5 μM/well oligomycin (Olig, port A), 1 μM/well FCCP (port B), and 0.5 μM/well rotenone/antimycin A (AA/Rot, port C). For the ATP production rate assay (Agilent, Santa Clara, CA, USA), the final concentrations used were 1.5 μM/well oligomycin (Olig, port A) and 0.5 μM/well rotenone/antimycin A (AA/Rot, port C). The OCR and ECAR values were normalized to the total number of cells per well. XFe Wave software 1.0.0-532 (Seahorse Bioscience, Agilent, Santa Clara, CA, USA) was used to analyze the results (Seahorse Bioscience, Agilent, Santa Clara, CA, USA).

### 4.15. Intracellular ATP Content

Intracellular ATP content was measured in each group of neuronal cells with the Intracellular ATP Assay Kit version 2 (TOYOIN<GROUP, Tokyo, Japan), according to the manufacturer’s specifications. After co-culturing in 24-well plates for 8 h, the cells in each group were washed twice with PBS, 400 μL ATP extraction reagent was added and stirred for 5 min at RT. 10 μL of ATP extraction suspension was mixed with 100 μL of ATP assay reagent. The luminescence of each group of cells was measured using the GloMax^®^ Discover microplate detector (Promega).

### 4.16. Intracellular Lactate Content

Intracellular lactate content was measured in each group of neuronal cells using the Lactate-Glo^TM^ Assay (Promega, Madison, WI, USA), according to the manufacturer’s specifications. After co-culturing in 24-well plates for 8 h, the cell supernatant from each group was collected and diluted 55-fold. The diluted supernatant was transferred (50 μL) to a 96-well plate, 50 μL of lactate detection reagent was added, and the plate was shaken for 45 s and then incubated for 60 min at RT. The luminescence of each group of cells was measured using the GloMax^®^ Discover microplate detector (Promega).

### 4.17. Cell Viability

Cell counting kit-8 (CCK-8; Dojindo, Kumamoto, Japan) was used to detect the viability of neurons in each group. Briefly, MELAS neurons were co-cultured with REC/MSC for 24 h. After incubating the collected group of neurons (Control-N, MELAS-N, MELAS-N w/REC, and MELAS-N w/MSC) in 96-well plates (10,000 cells per well) for 1 week at 37 °C, then the groups of cells were incubated with CCK-8 solution for 2 h at 37 °C. Absorbance was measured at 450 nm using a GloMax^®^ Discover microplate detector (Promega, Madison, WI, USA).

### 4.18. Detection of Human Growth/Differentiation Factor 15 (GDF-15) Levels

Cell culture medium samples were prepared according to the manufacturer’s protocol, and the levels of GDF-15 were measured using an enzyme-linked immunosorbent assay (ELISA) kit (Invitrogen, Carlsbad, CA, USA). The optical density of the samples was measured using a GloMax^®^ Discover Microplate Reader (Promega, WI, USA) at 450 nm.

### 4.19. Statistical Analysis

Data were analyzed, and statistical analyses were performed using GraphPad Prism 9 (GraphPad Software, San Diego, CA, USA). Data points are expressed as mean ± SD unless otherwise indicated. Statistical analyses were performed by one-way ANOVA with a Bonferroni post hoc analysis for comparison of three or more groups. For comparisons between groups, Student’s t-test was used. Results with * *p* < 0.05, ** *p* < 0.01, *** *p* < 0.001, and **** *p* < 0.0001 were considered significant.

## 5. Conclusions

These data suggest that REC-donated exogenous mitochondria may offer a potential therapeutic strategy for treating neurological dysfunction in MELAS.

## Figures and Tables

**Figure 1 ijms-24-17186-f001:**
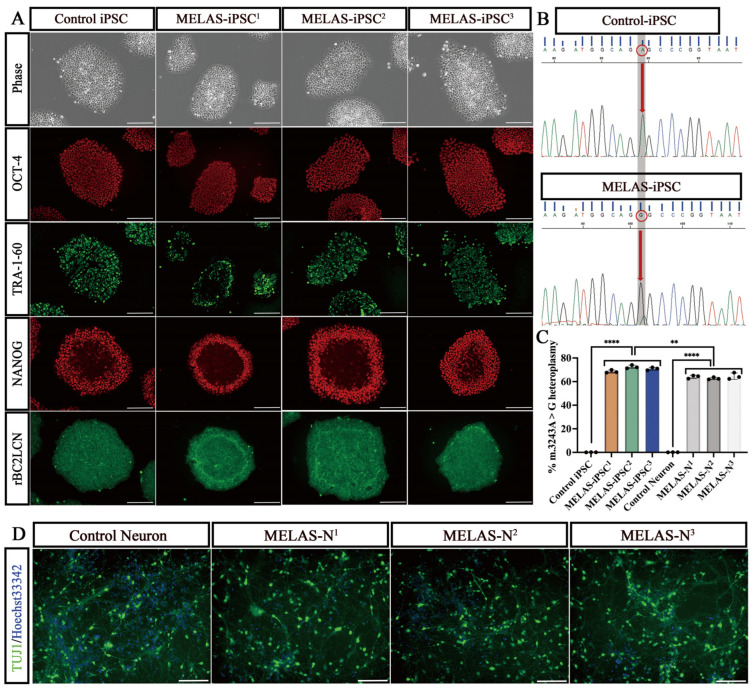
Heteroplasmy of iPSC-induced neurons. (**A**) Immunostaining of pluripotency and surface markers (OCT4, TRA-1-60, NANOG, and rBC2LCN) in representative iPSCs (Control-iPSC, MELAS-iPSC^1^, MELAS-iPSC^2^, and MELAS-iPSC^3^). Scale bars, 400 μm. (**B**) Sequencing chromatogram showing the heteroplasmic m.3243A>G mutation in iPSCs from patients with MELAS. Red circles represent adenine (A)-guanine (G) transitions on mtDNA 3243 nucleotides. (**C**) Quantification of the percentage of m.3243A>G heteroplasmy in iPSCs and their induced neurons in each group (*n* = 3). (**D**) Immunostaining for the surface marker TUJ1 in representative neurons (control neurons, MELAS-N^1^, MELAS-N^2^, and MELAS-N^3^). Scale bars, 100 μm. Data represent the mean ± standard deviation (SD) of three independent experiments. ** *p* < 0.01, **** *p* < 0.0001.

**Figure 2 ijms-24-17186-f002:**
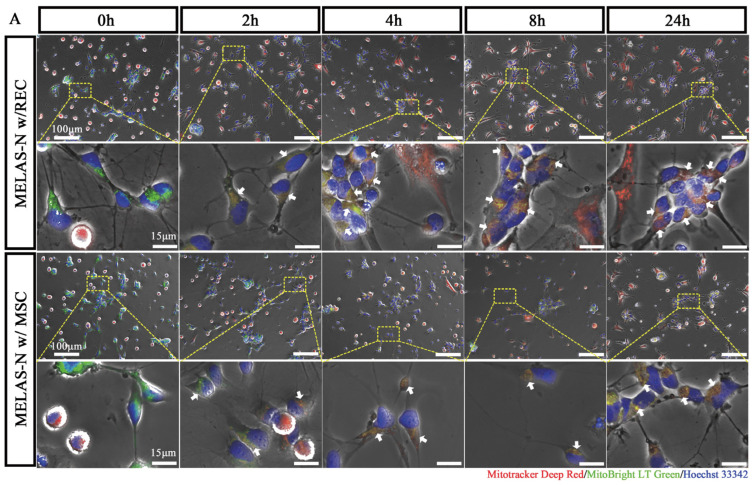
Mitochondrial transfer between RECs and neurons. The contact and non-contact co-culture systems are shown in (**B**). (**A**) Representative fluorescence microscopy images of MELAS neurons (MELAS-N) and REC/MSCs in the contact co-culture system at 0, 2, 4, 8, and 12 h. White arrows: Red fluorescence of mitochondria from MSCs/RECs was expressed in neurons containing green fluorescence, indicating successful mitochondrial transfer. (**C**) Representative images of MELAS neurons and REC/MSC in a non-contact co-culture system (0.4 μm or 3 μm cell culture insert). (**D**) Representative distribution of DiO-labeled MELAS neurons and Mito-Red-labeled MSCs/RECs in co-cultured cells analyzed by flow cytometry. (**E**) Time course of the mitochondrial transfer rate between Q2 phase-distributed MELAS neurons and RECs/MSCs is presented, as well as the percentage of this population over the total MELAS neurons (*n* = 4). (**F**) Quantification of the mitochondrial transfer rate of MSCs/RECs to MELAS neurons in different somatic co-culture systems (*n* = 6). Data represent the mean ± standard deviation (SD) of three independent experiments. ns, not significant. * *p* < 0.05; ** *p* < 0.01; **** *p* < 0.0001.

**Figure 3 ijms-24-17186-f003:**
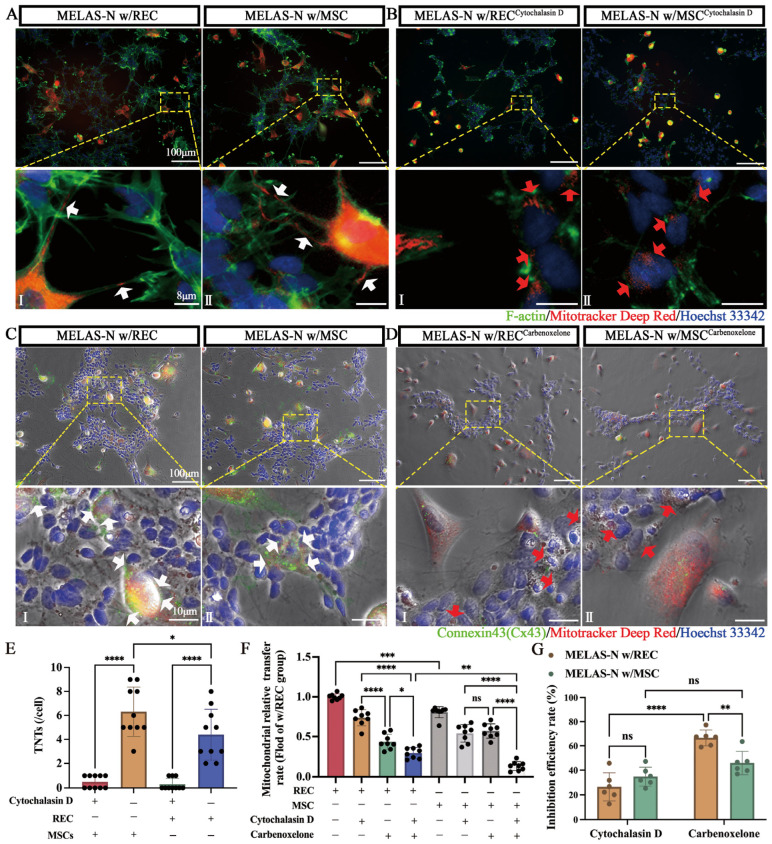
MSCs/RECs connect with MELAS neurons (MELAS-N) via TNTs and Cx43-containing GJCs. (**A**) Representative fluorescence microscopy images of MELAS neurons co-cultured with RECs (I) or MSCs (II). (**B**) Representative images of MELAS neurons co-cultured with RECs (I) or MSCs (II) after the addition of cytochalasin D. Mito-Red labeled RECs or MSCs, Hoechst33342 labeled MELAS neurons, and F-actin labeled all the cells. White arrowheads: REC/MSC mitochondria in TNTs. Red arrows: REC/MSC mitochondria were present in MELAS neurons, even without TNT formation. (**C**) Representative immunofluorescence images of MELAS neurons co-cultured with RECs (I) or MSCs (II). (**D**) Representative images of MELAS neurons co-cultured with REC (I) or MSC (II) after the addition of carbenoxelone. Mito-Red labeled REC or MSC, Hoechst33342 labeled MELAS neurons, and Connexin43 labeled all cells. White arrows indicate high Cx43 expression at the cell junctions. Red arrows indicate reduced Cx43 expression after gap junction inhibition, but REC/MSC mitochondria are still present in the MELAS neurons. (**E**) The number of TNTs in cells from each group was calculated with or without cytochalasin D (*n* = 10). (**F**) Quantification of mitochondrial transfer rate of RECs/MSCs to MELAS neurons in response to inhibitors (cytochalasin D and carbenoxelone) (*n* = 8). (**G**) Inhibitory efficacy of different inhibitors on REC/MSC mitochondrial transfer rate (*n* = 6). Data represent the mean ± standard deviation (SD) of three independent experiments. ns, not significant. * *p* < 0.05; ** *p* < 0.01; *** *p* < 0.001; **** *p* < 0.0001.

**Figure 4 ijms-24-17186-f004:**
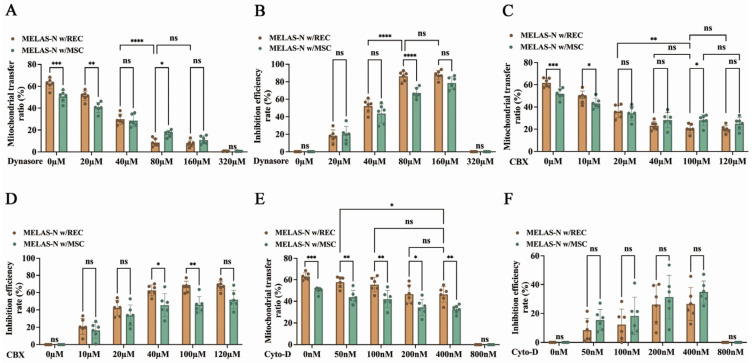
Mitochondrial transfer rates of RECs and MSCs to MELAS neurons (MELAS-N) under various inhibitors. Mitochondrial transfer rates of RECs and MSCs to MELAS neurons at different inhibitor (dynasore (**A**), carbenoxelone (**C**), and cytochalasin D (**E**)) concentration gradients (*n* = 6). Inhibition of REC and MSC mitochondrial transfer using different concentrations of gradient inhibitors (dynasore (**B**), carbenoxelone (**D**), and cytochalasin D (**F**)) (*n* = 6). (**G**) RECs/MSCs donate mitochondria via multiple mechanisms (*n* = 6). (**H**) Schematic diagram of the 3 μm non-contact system (with or without dynasore/carbenoxelone, cytochalasin D). (**I**) Mitochondrial transfer rate of REC in 3 μm non-contact system with different inhibitors (*n* = 6). Data represent the mean ± standard deviation (SD) of three independent experiments. ns, not significant. * *p* < 0.05; ** *p* < 0.01; *** *p* < 0.001; **** *p* < 0.0001.

**Figure 5 ijms-24-17186-f005:**
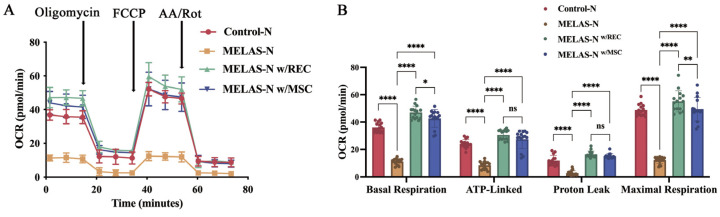
Recovery of cellular respiration activity in MELAS neurons receiving REC mitochondria. (**A**) Oxygen consumption rate (OCR) of neurons in each group (control neurons, MELAS neurons, MELAS neurons co-cultured with REC, and MELAS neurons co-cultured with MSC) was measured over time (min). The order of injection of oligomycin, FCCP, and AA/Rot is shown. (**B**) Basal respiration, ATP production, proton leakage, and maximum respiration of the neurons in each group were calculated (*n* = 15). (**C**) OCR of neurons in each group (control neurons, MELAS neurons, MELAS neurons co-cultured with normal REC mitochondria, and MELAS neurons co-cultured with REC mitochondria (Rot-treated)) was measured over time (min). (**D**) Basal respiration, ATP production, proton leakage, and maximum respiration were calculated for each group (*n* = 15). Data represent the mean ± standard deviation (SD) of three independent experiments. ns, not significant. * *p* < 0.05; ** *p* < 0.01; **** *p* < 0.0001.

**Figure 6 ijms-24-17186-f006:**
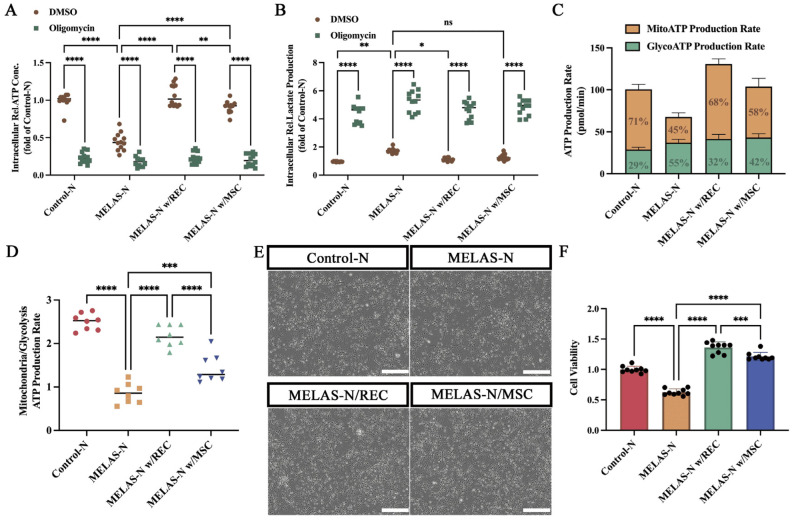
Recovery of bioenergetic in MELAS neurons. (**A**) Intracellular ATP content was determined in each group after oligomycin treatment (*n* = 12). (**B**) Intracellular lactate levels were determined in each group after oligomycin treatment (*n* = 12). (**C**) Seahorse XFp metabolic flux analysis of mitochondrial ATP production rates and glycolytic ATP production rates in each group of neurons (*n* = 8). (**D**) XFp ATP rate index calculated from data in panel C (*n* = 8). (**E**) Representative images of neurons from each group. Neurons in the co-culture group were co-cultured with REC/MSCs for 24 h; neurons from each group were collected and seeded into 96-well plates, cultured for 1 week, and then tested. Scale bars, 100 μm. (**F**) The CCK-8 assay was used to detect viability in each group (*n* = 9). Data represent the mean ± standard deviation (SD) of three independent experiments. ns, not significant. * *p* < 0.05; ** *p* < 0.01; *** *p* < 0.001; **** *p* < 0.0001.

**Figure 7 ijms-24-17186-f007:**
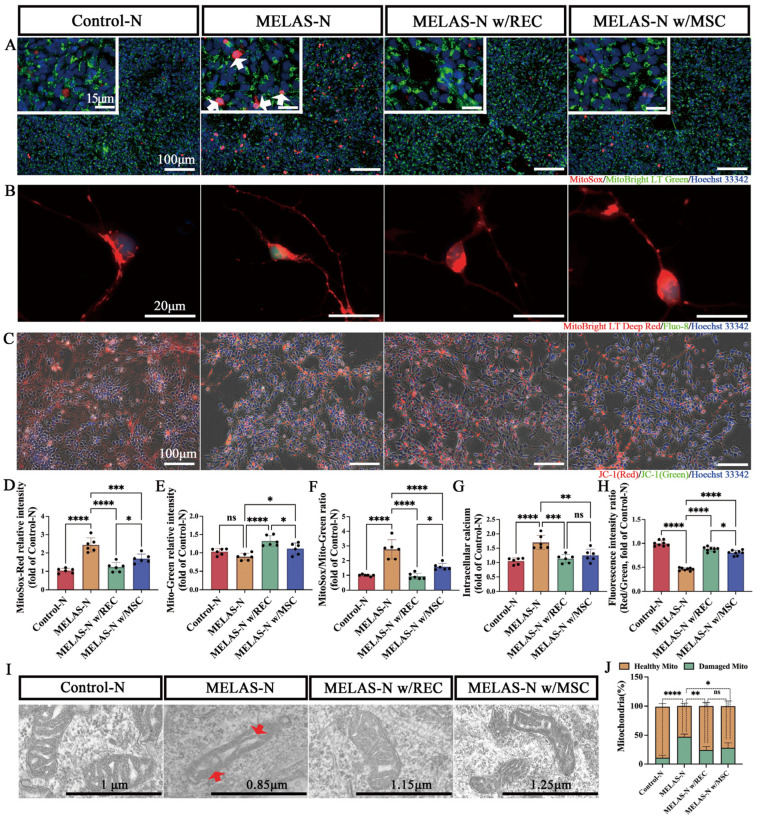
Recovery of mitochondrial function in MELAS neurons after co-culture in a non-contact system. (**A**) Representative fluorescence microscopy images of MitoSOX and MitoBright LT green staining in each neuronal group. White arrows indicate high intensity MitoSox dyes. (**B**) Representative images of fluo-8 (calcium-sensitive fluorescent) staining in each group of neurons. (**C**) Representative images of JC-1 (response mitochondrial membrane potential, MMP) staining in each group of neurons. (**D**) Quantification of ROS levels by analyzing the fluorescence intensity of MitoSox (*n* = 6). (**E**) Quantification of mitochondrial levels by analyzing the fluorescence intensity of MitoBright LT green (*n* = 6). (**F**) Quantification of mitochondrial ROS in neurons (*n* = 6). (**G**) Quantification of intracellular calcium levels by analyzing the fluorescence intensity of fluo-8 (*n* = 6). (**H**) Fluorescence intensity ratio of MMP levels in groups of neurons (*n* = 8). (**I**) Mitochondrial morphology of the neurons in each group. Red arrows indicate the absence and breakage of mitochondrial cristae. (**J**) Quantification of damaged mitochondria in different groups with percentage of mitochondria showing abnormal cristae. More than 50 mitochondria in each group (52 in Control-N, 50 in MELAS-N, 55 in MELAS-N w/REC, and 51 in MELAS-N w/MSC) were examined. Data represent the mean ± standard deviation (SD) of three independent experiments. ns, not significant. * *p* < 0.05; ** *p* < 0.01; *** *p* < 0.001; **** *p* < 0.0001.

**Figure 8 ijms-24-17186-f008:**
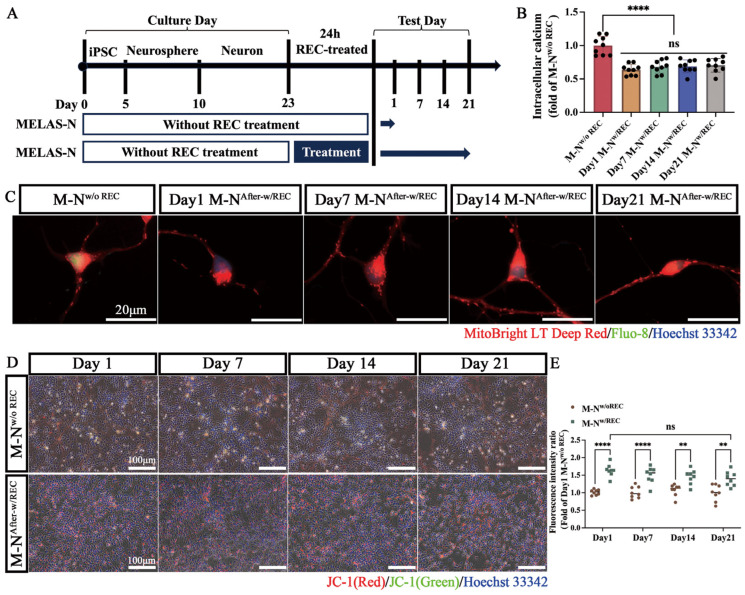
Persistency of REC functional mitochondria. (**A**) Schematic representation of the experiment used to assess the duration of REC mitochondrial function. (**B**) Intracellular calcium levels were quantified in MELAS neurons (MELAS-N) grown on days 1, 7, 14, and 21 (after receiving REC-donated mitochondria) and in MELAS neurons with high heteroplasmy (*n* = 9). (**C**) Representative fluorescence microscopy images of fluo-8 staining in each group of MELAS neurons on days 1, 7, 14, and 21. (**D**) Representative images of MELAS and REC-treated MELAS neurons stained with JC-1 on days 1, 7, 14, and 21. (**E**) Quantification of JC-1 fluorescence intensity ratios in REC-treated and untreated MELAS neurons (*n* = 8). (**F**) Expression levels of GDF-15 in different groups of neuronal culture media (*n* = 7). (**G**) GDF-15 levels were quantified in MELAS neurons grown on days 1, 7, 14, and 21 (after receiving REC-donated mitochondria) and in MELAS neurons with high heteroplasmy (*n* = 7). (**H**) Seahorse XFp metabolic flux analysis of mitochondrial ATP production rates and glycolytic ATP production rates in each group of MELAS neurons (*n* = 8). (**I**) XFp ATP rate index calculated from data in panel H (*n* = 8). Data represent the mean ± standard deviation (SD) of three independent experiments. ns, not significant. ** *p* < 0.01; **** *p* < 0.0001.

**Figure 9 ijms-24-17186-f009:**
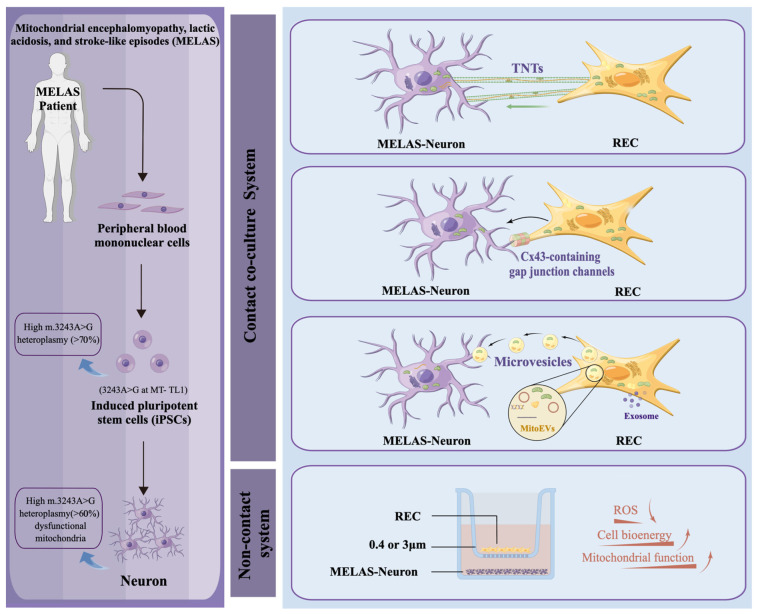
Schematic diagram of exogenous mitochondrial multiple pathways regulating functions related to MELAS neuronal regulation. iPSCs from patients with MELAS with high levels of heteroplasmy (m.3243A>G mutation) were used to establish a MELAS neuron model and to evaluate the ability of REC to donate mitochondria under the classical mitochondrial transfer pathway (TNTs, gap junctions, MDVs, and MVs) and the functionality of such exogenous mitochondria. This figure was made using Figdraw.

## Data Availability

The datasets used and/or analyzed during the current study are available from the corresponding author upon reasonable request.

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
