# Peer review of "MELAS-Derived Neurons Functionally Improve by Mitochondrial Transfer from Highly Purified Mesenchymal Stem Cells (REC)"

_ijms, 2023, doi:10.3390/ijms242417186_

Round 1
Reviewer 1 Report
Comments and Suggestions for Authors
A study using induced pluripotent stem cells shows that highly purified mesenchymal stem cells (RECs) can efficiently donate mitochondria to iPSC-induced MELAS neurons, outperforming mesenchymal stem cells. This mitochondrial transfer significantly restores mitochondrial function and supports neuronal survival.
The study follows on from results previously published by the same authors (DOI:10.1186/s13287-023-03274-y). Comparing different aspects of mitochondrial transfer from MSCs and RECs, the story of the manuscript is logical with a clear link. I found it interesting that the transferred mitochondria persisted in iPS-derived neurons for 21 days. Does the intensity of the MitoTracker staining remain at a similar level during three weeks of cultivation? What were the culture conditions for the neurons? What percentage of the neurons died during these three weeks?
I would recommend that you give individual points in the results obtained, not just bar graphs. Otherwise, I have no comments.
Reviewer 2 Report
Comments and Suggestions for Authors
In the manuscript by Liu et al. the authors show how transferred mitochondria from MSCs can restore bioenergetics of mtDNA deficient neurons. While the study is interesting, the study design appropriate and the data is mostly convincing and supports the conclusions, I have some concerns that should be addressed before the paper can be accepted for publication.
First of all in the figure 3 it is very difficult, if not impossible, to see the tunnelling nanotubes or the mitochondria in them, higher magnification and resolution images would be needed to convince the reader that the data is solid.
In the discussion the authors state that the morphology of the melas neurons is normal for mature neurons. After short, 3 week, differentiation, there is no way that the cells are mature neurons. Already the fact that the cells proliferate tells that they are not mature as mature neurons are postmitotic and do not proliferate. This also relates to the viability results in fig 6e-f. Analysing proliferation does not necessarily tell about the viability of the cells, it may also relate to the differentiation status of the cells, as neurons will stop proliferating when they mature and proliferate at different rates at different cell stages.
The ROS results are very strange. How do the authors explain the reduced ROS levels in the treated cells. Transferring more functional mitochondria to the cells should rather increase ROS production than reduce it. I understand that the melas neuronshave a higher ROS level than control cells, even if they respirate less, but how does adding more mitos and inducing more respiration and increasing oxygen consumption decrease ROS production? This is very controversial.
In figure 7 the authors show that the coculture improves mitochondrial morphology. But do they know if they are looking at the original host cell mitos or the transferred donor mitos?
Minor comments:
Page 2 intro the sentence "The polyploid nature and replication segregation hamper of mtDNA ..." is strange
Page 4 "not through mitochondria" do the authors mean not permeable to mitochondria
Comments on the Quality of English Language
There are some sentences that are strange and need clarification but overall the English language is OK.
